# Determinants of exit-knowledge of ambulatory patients on their dispensed medications: The case in the outpatient pharmacy of Mizan-Tepi University Teaching Hospital, Southwest Ethiopia

**Semere Welday Kahssay**[1]*, **Peacock Mulugeta**[2]

**1** Department of Pharmaceutical Chemistry, School of Pharmacy, College of Medicine and Health Sciences, Mizan-Tepi University, Mizan-Aman, Ethiopia, **2** School of Pharmacy, College of Medicine and Health Sciences, Mizan-Tepi University, Mizan-Aman, Ethiopia

\* semere0409@gmail.com

## Abstract

### Background

Patient's knowledge about dispensed medications is one of the major factors that determine the rational use of medicines.

### Objectives

This study aimed to assess exit-knowledge of ambulatory patients about their dispensed medications and associated factors at the outpatient pharmacy of Mizan-Tepi University Teaching Hospital, Southwest Ethiopia.

### Methods

A hospital-based cross-sectional study design was conducted from August to October 2021. Study subjects were selected by random sampling technique and were interviewed using a structured interview questionnaire. Binary logistic regression was used to identify factors associated with exit knowledge. At a 95% confidence interval (CI), $p \leq 0.05$ was considered statistically significant.

### Result

Of the total 400 participants, 116 (29.0%) participants had sufficient exit-knowledge about their dispensed medication. Patients with higher educational level had increased exit knowledge of dispensed medications than those with no formal education (AOR: 5.590; 95% CI 1.019–30.666). Also, the nature of illness as being chronic significantly enlarged the odds (AOR 5.807; 95% CI 2.965–11.372) of having sufficient exit-knowledge. Participants who reported, "I do not know" and "I did not get enough information from the pharmacist" had lower odds (AOR 0.374; 95% CI: 0.142–0.982) and (AOR 0.166; 95% CI 0.062–0.445) of

**Data Availability Statement:** All relevant data are within the paper and its Supporting Information files.

**Funding:** The authors received no specific funding for this work.

**Competing interests:** The authors have declared that no competing interests exist.

sufficient exit-knowledge in comparison to those who responded "I got enough information from the pharmacist" respectively. Furthermore, the odd of sufficient exit-knowledge was 7.62 times higher in those who claimed prescribing doctor as the source of information.

## Conclusion

The majority of patients had insufficient exit-knowledge about their dispensed medications. Educational status, nature of the disease, perceived sufficiency of pharmacist knowledge, and source of information were significantly associated with exit knowledge.

## Introduction

One major factor that affects the rational use of medicines by patients is the practice of dispensing. Pharmacy professionals are in charge of good dispensing practice which refers to the provision of the right drug in the right dosage, quantity, and package to the right patient with clear instruction and appropriate follow-up [1–3]. Dealing with factors that affect the dispensing practice of pharmacists such as communication barriers, dispensing premises, the high workload of dispensers, and Pharmacist's skill, knowledge, and commitment are essential to ensure patients' sufficient knowledge about their dispensed medications [4, 5].

If patients fail to get sufficient information regarding their medications, it can result in a decrease in efficacy of medication, treatment failure, development of adverse drug reactions, and an increase in the risk of drug resistance [6, 7]. A strong pharmacist-patient interaction and multidisciplinary collaboration are crucial to assure the safe and effective use of a drug, and decrease the existence of drug therapy problems [8, 9].

Though irrational drug dispensing is worsened in developing countries, it becomes a worldwide problem. Ethiopia is no exception, irrational drug prescribing and dispensing, and inappropriate uses of medicines by patients are common. Dispensing of medications with wrong dosage and duration, poor/without labeling, absence of patient counseling, incorrect drug information, and poor patients understanding and knowledge of their medication dosage are a few of the irrational dispensing practices noticed in Ethiopia [2, 10, 11].

Despite the severity of the problem, there is no evidence of exit knowledge of ambulatory patients for dispensed drugs in South West Region of Ethiopia. Therefore, this study is aimed to determine the exit knowledge of ambulatory patients, and the effect of patient-pharmacist communication and other factors on patients' exit knowledge of their dispensed medications. The findings of this study would promote good dispensing and counseling practices, and thus encourage safe and cost-effective use of drugs.

## Patients and methods

### Study design, period, and setting

A hospital-based cross-sectional study was conducted among ambulatory patients visiting the outpatient pharmacy of Mizan-Tepi University Teaching Hospital (MTUTH) from August to October 2021. MTUTH is a public referral and teaching hospital found in Mizan-Aman town, Bench-Sheko zone of Southwest Ethiopia. Mizan-Aman town is found 568 km southwest of Addis Ababa, the capital city of Ethiopia. The hospital serves the people of Mizan-Aman town and the surrounding community with a catchment population of more than 2.75 million people [12]. The hospital provides general in-patient, outpatient, and intensive care unit services

in its major departments (internal medicine, surgery, pediatrics, and gynecology) and other units.

## Study population

Those ambulatory patients who were mentally stable, willing to participate, and visited the outpatient pharmacy during the study period were included in the study. On the contrary, patients with poly-pharmacy (patients with more than three dispensed medications) were excluded from the study because this can potentially lead to increased difficulty in information retention and recalling ability.

## Sample size and sampling technique

The sample size was determined using the single population proportion formula as follows:

$$n = \frac{(Z\,\alpha_{1/2})^2 \times P\,(1-P)}{d^2} = 364$$

Where, $n$ = sample size from the formula; $Z\alpha_{1/2}$ = 1.96 (standard Z value for a two-sided 95% confidence interval), $P$ = 0.386 (38.6% sufficient exit knowledge reported from the study conducted in Hiwot Fana specialized University Hospital) [13], and d = absolute precision (at 5%, d = 0.05). Adding 10% for nonresponse (364*0.1 = 36.4), the final estimated sample size was calculated to be 400. By using a simple random sampling method eligible participants were consecutively enrolled until the required sample size was reached.

## Data collection techniques and procedures

A comprehensive data collection tool was developed by adopting previously conducted studies [13–16]. It consisted of three major sections; the first section was designed to assess the sociodemographic characteristics of the participants. The second and third part was prepared to determine the perception of patients towards pharmacists' service, and the exit- knowledge of participants about their dispensed medications, respectively. To determine the exit-knowledge level, the key outcome variable, 15 different questions were developed. The participant was considered to have sufficient exit-knowledge about the dispensed medication when he/she responded a positive answer for at least two-thirds (≥10 out of 15) of the total knowledge assessing questions. Those who gave a positive response for less than two-thirds (≤9 out of 15) of the total knowledge assessing questions were considered as having insufficient exit-knowledge [14–16]. The collection of data was carried out by two senior pharmacists with the daily supervision of the principal investigator.

## Data quality control, analysis and interpretation

The questionnaire was first translated to local language, Benchigna, to keep it nuanced. To ensure the quality of data, a pretest was conducted on 5% of the study population, and some corrections were made based on the results obtained from the pretest. The questionnaire was checked for proper filling on a daily basis instantly at the time of data collection.

The Statistical Package for the Social Science (SPSS) version 24 was used for the analysis of cleaned data. The frequency and percentage of the variables were determined by using descriptive statistics. Univariate logistic regression followed by binary logistic regression was employed to identify factors associated with exit-knowledge of patients about their dispensed medications. Potential predictor variables with a p-value ≤0.25 in univariate analysis were considered for subsequent binary logistic regression analysis. Finally, predictor variables with

**Table 1. Socio-demographic characteristics of ambulatory patients attending Mizan-Tepi University Teaching Hospital from August to October 2021.**

| Variables | Frequency (%) |
|---|---|
| Sex | |
| Male | 239 (59.8) |
| Female | 161 (40.3) |
| Age | |
| 18–39 | 305 (76.3) |
| 40–59 | 80 (20.0) |
| >60 | 15 (3.8) |
| Religion | |
| Orthodox | 131 (32.8) |
| Protestant | 168 (42.0) |
| Muslim | 81 (20.3) |
| Catholic | 20 (5.0) |
| Marital status | |
| Single | 148 (37.0) |
| Married | 212(53.0) |
| Divorced | 22(5.5) |
| Widowed | 18(4.5) |
| Educational status | |
| Illiterate | 36 (9.0) |
| Read and write | 49 (12.3) |
| Primary | 97 (24.3) |
| Secondary | 121(30.3) |
| Tertiary | 97 (24.3) |
| Occupation | |
| Farmer | 90 (22.5) |
| Government employee | 59 (14.8) |
| Merchant | 75 (18.8) |
| Private employee | 27 (6.8) |
| Student | 75 (18.8) |
| House wife | 51 (12.8) |
| Retired | 17 (4.3) |
| Unemployed | 6 (1.5) |
| Ethnicity | |
| Bench | 131 (32.8) |
| Keffa | 83 (20.8) |
| Oromo | 57 (14.3) |
| Amhara | 84 (21.0) |
| Tigray | 18 (4.5) |
| Sheka | 11 (2.8) |
| Others* | 16 (4.0) |
| Residence | |
| Urban | 279 (69.8) |
| Rural | 121 (30.3) |
| Monthly Income | |
| 2000 and below | 253(63.3) |
| 2001–6000 | 133(33.3) |

(*Continued*)

**Table 1.** (Continued)

| Variables | Frequency (%) |
|---|---|
| 6001–10,000 | 12(3.0) |
| 10,001 and above | 2(0.5) |

Asterisk (*) stands for Gurage, Wolayita, Silte, Dawro, and Sidama.

a p-value ≤0.05 in binary logistic regression were considered to have a significant association with the outcome variable. Graphpad prism version 9.1.2 was used for drawing graph.

## Ethical considerations

Ethical permission was obtained from the Research and Ethical Review Committee of Mizan-Tepi University (MTU/1621/2013). The purpose of the study was explained to the study subjects and oral informed consent was taken prior to data collection. The participants were informed that they had the right to decline to participate or withdraw from the study at any time. To assure the anonymity of the respondents, any personally identifiable information was avoided in the questionnaire and the information provided by the participants was kept confidential.

## Result

### Socio-demographic characteristics of study participants

From a total of 400 participants who were involved in the study, more than half were males (239, 59.8%) and the majority of them (187, 46.8%) ranged from 18 to 39 years. Religious-wise, most of the participants were Protestants (168, 42.0%). Regarding marital status, more than half of the study subjects (212, 53.0%) were married. A significant proportion of the respondents (121, 30.3%) completed secondary school, followed by primary school (97, 24.3%). The majority of the study subjects were farmers (90, 22.5%), followed by merchants, & students (75, 18.8%). Most of the participants (131, 32.8%) belong to the Bench ethnic group, and (279, 69.8%) were urban residents (Table 1).

### Patient- pharmacist interaction

As depicted in Table 2, for most of the participants (161, 40.3%), it is their first visit in the last six months, and a large proportion of the study subjects (235, 58.8%) came for acute illness. The main media of communication with the pharmacists was Benchigna (132, 33.0%). More than half of the participants (236, 59.0%) disclosed that they had good interaction with the pharmacy personnel. A similar proportion of the respondents reported that the dispensers were polite. Furthermore, most of them (329, 82.3%) revealed that the voice and tone of the pharmacists were clear, and (256, 64.0%) of the participants stated that the information provided on how to take their medications was understandable. About half of the participants perceived the waiting area as not comfortable. The preponderance of the study subjects (254, 63.5%) perceived the sufficiency of the dispenser's information as enough.

### Knowledge status of the patients

Around one-third of the study subjects (129, 32.3%) recalled the name, and more than three-fourths (318, 79.5%) recalled the indication, (318, 79.5%) duration of treatment, and (346, 86.5%) frequency of each medication. Almost all of the respondents recalled the route of

**Table 2. Patients' perception of the outpatient pharmacy service of Mizan-Tepi University Teaching Hospital from August to October 2021.**

| Variables | Frequency (%) |
|---|---|
| Frequency of dispensary unit visit (6 month) | |
| First time | 161 (40.3) |
| Second time | 82 (20.8) |
| Repeated times | 157 (39.3) |
| Perception of nature of the disease | |
| Acute | 235 (58.8) |
| Chronic | 165 (41.3) |
| Primary language of communication | |
| Bench | 132 (33.0) |
| Keffa | 74 (18.5) |
| Amharic | 113 (28.5) |
| Afaan Oromo | 45 (11.3) |
| Tigregna | 13 (3.3) |
| Sheko | 10 (2.5) |
| Other | 13 (3.3) |
| Perceived interaction status with pharmacist rated by the patient | |
| Good | 236 (59.0) |
| Moderate | 108 (27.0) |
| Poor | 56 (14.0) |
| Perceived clearness of pharmacist's voice & tone | |
| Clear | 329 (82.3) |
| Not clear | 71(17.8) |
| Perceived comfort of the waiting area | |
| Very comfortable | 4 (1.0) |
| Comfortable | 36 (9.0) |
| Fairly comfortable | 51 (12.8) |
| Not comfortable | 201 (50.3) |
| Very uncomfortable | 74 (18.5) |
| Not available | 34 (8.5) |
| Perceived politeness of service providers | |
| Very polite | 44 (11.0) |
| Polite | 237 (59.3) |
| Fairly polite | 89 (22.3) |
| Impolite | 30 (7.5) |
| Perceived clearness of pharmacist's instruction for patient | |
| Clear | 256 (64) |
| Fairly clear | 98 (24.5) |
| Not clear | 46 (11.5) |
| Perceived sufficiency of pharmacist's information | |
| Enough | 254 (63.5) |
| Not enough | 82 (20.5) |
| I do not know | 64 (16.0) |

administration (387, 96.8%). Nearly all of the participants (373, 93.3%) and more than half of the participants (260, 65.0%) did not know the major possible side-effects, and what to avoid while taking their medications, respectively. Furthermore, around two-thirds of the study

subjects (259, 64.8%) were aware of the proper storage conditions of their medications. Labeling on medications was made only for (137, 34.25%) participants; of this (62, 45.25%) understood the labeling. Overall, the majority of the respondents (284, 71.0%) had insufficient exit knowledge of their dispensed medications (Table 3).

### Sources of information and number of drugs received

As depicted in Fig 1, the majority of the respondents (296, 74%) obtained information regarding their medications from dispensing pharmacists. A significant proportion of the participants (32, 8%) witnessed that they didn't get any information about the prescribed medications from anybody.

Among study subjects about half (51.1%) of them received two drugs, 146 (36.5%) received only one drug; the rest 12% of the participants received three drugs.

### Factors affecting exit-knowledge of the patients

As it can be seen from Table 4, Exit-knowledge of the patients about their dispensed medication was found to have a significant association with several predictor variables. Educational status, nature of the disease, perceived sufficiency of pharmacist's information, and source of information were the predictor variables that significantly affected patients' level of exit-knowledge.

Participants who completed higher (tertiary) education were 5.59 times more likely to have sufficient exit knowledge of dispensed medication than those who did not have formal education (AOR: 5.590; 95% CI 1.019–30.666).

Nature of illness as being chronic significantly enlarged the odds (AOR 5.807; 95% CI 2.965–11.372) of having sufficient exit-knowledge than acute. Our study also indicates that, participants who reported: "I do not know" and "I did not get enough information from the pharmacist" had lower odds (AOR 0.374; 95% CI: 0.142–0.982) and (AOR 0.166; 95% CI 0.062–0.445) of sufficient exit-knowledge in comparison to those who responded "I got enough information from the pharmacist", respectively. Furthermore, the odd of sufficient exit-knowledge was 7.620 times higher in those who claimed prescribing doctor as the source of information as compared to those who said dispensing pharmacist (Table 4).

## Discussion

In this prospective cross-sectional study, the outpatient pharmacy service, exit-knowledge of patients on their dispensed medications, and associated factors among those who attended the outpatient pharmacy of MTUTH were investigated.

The finding of this study revealed that the majority of the patients (79.5%, 96.8%, 86.5%, 79.5%, 77.3%, 64.8%, and 81.0%) recalled the indication, route, frequency, duration, instruction on how to use, proper storage, and the expected therapeutic outcomes of dispensed medication's, respectively. This finding is in line with studies conducted in Ayder Comprehensive Specialized Hospital (ACSH), Mekelle [14]; Hiwot Fana Specialized University Hospital (HFSUH), Harar [13]; Chencha Primary Hospital [17], and Ambo General Hospital (AGH), Ambo [16]. This shows that in the aforementioned aspects of drugs, most ambulatory patients had sufficient exit knowledge. On the flip side, our study pointed out that only a lower proportion of the participants (32.3%, 16.0%, 6.8%, 35.0%, 17.8%, and 41.8%) knew the name, drug interactions, major side effects, what to avoid, what to do when an adverse reaction happens and lifestyle modification to undertake for dispensed medications, respectively. This finding is also consistent with report of the forementioned studies. This might be due to either patient related factors (like being reluctant to know about their medication) or health care provider

**Table 3. Exit Knowledge status of ambulatory patients attending Mizan Tepi University Teaching Hospital from August to October 2021.**

| Variables | Frequency (%) |
|---|---|
| know the name of the dispensed medications | |
| Yes | 129 (32.3) |
| No | 271 (67.8) |
| know reasons (indication) for taking medications | |
| Yes | 318 (79.5) |
| No | 82 (20.5) |
| know about the route of administration of dispensed medications | |
| Yes | 387 (96.8) |
| No | 13 (3.3) |
| recall the frequency of dispensed medications | |
| Yes | 346 (86.5) |
| No | 54 (13.5) |
| recall about the duration of therapy | |
| Yes | 318 (79.5) |
| No | 82 (20.5) |
| understand labeling on medications | |
| Yes | 62 (45.25) |
| No | 75 (54.75) |
| Was not labeled | 263 (65.8) |
| know about any drug interactions (drug-drug or drug-food or drug-disease interactions) | |
| Yes | 64 (16.0) |
| No | 336 (84.0) |
| know about the major side effects of dispensed medications | |
| Yes | 27 (6.8) |
| No | 373 (93.3) |
| know instructions on how to use medication | |
| Yes | 309 (77.3) |
| No | 91 (22.8) |
| know what to do in cases of missed dose | |
| Yes | 228 (57.0) |
| No | 172 (43.0) |
| know how to store medications properly | |
| Yes | 259 (64.8) |
| No | 141 (35.3) |
| know what to avoid during taking the medication | |
| Yes | 140 (35.0) |
| No | 260 (65.0) |
| know what to do if adverse reactions happen | |
| Yes | 71 (17.8) |
| No | 329 (82.3) |
| know about any lifestyle modifications should be undertaken | |
| Yes | 167 (41.8) |
| No | 233 (58.3) |
| know about the expected therapeutic outcomes | |
| Yes | 324 (81.0) |
| No | 76 (19.0) |

*(Continued)*

**Table 3.** (Continued)

| Variables | Frequency (%) |
|---|---|
| Exit knowledge status of patients about dispensed medication | |
| Sufficient (at least two-thirds of correct answers) | 116 (29.0) |
| Not sufficient (less than two-thirds of correct answers) | 284 (71.0) |

related factors (like poor behaviour/ attitude, and high workload) or both [18]. Therefore to improve adherence and therapeutic outcomes, dispensers, and other healthcare providers should counsel patients and make sure they understood these aspects of drugs.

Our study revealed that 29.0% of participants had sufficient exit-knowledge regarding their dispensed medications. Though the study setting is different (community pharmacy), this result is consistent with the study done in Ghana in which the proportion of the study subjects who had adequate knowledge of dispensed medications was 31% [19]. But, the exit knowledge sufficiency of our respondents is less than the studies conducted in the AGH (55.5%) [16], ACSH, Mekelle (81%) [14], Gondar (38.3%) [20], Federal Harar Police Hospital (FHPH) (38.6%) [15] and HFSUH, Harar (46%) [13]. But, it is greater than studies done in Chencha Primary Hospital (13.2%) [17] and Gambia, Western Africa (16.1%) [21]. These discrepancies in the exit-knowledge of the participants could be attributed to differences in the calculation of knowledge level, percentage of labeled medications, frequency of pharmacy visit, number of drugs received, educational status of the patients, and study setting.

One of the predictor variables that significantly affected the exit-knowledge of ambulatory patients in our study setting was their educational level. Which pointed out that the level of exit-knowledge on dispensed medications was increased among participants who attended tertiary education compared to those without formal education (AOR: 5.590; 95% CI 1.019–30.666). A study conducted at HFSUH in Eastern Ethiopia revealed that respondents' educational level (higher education) increased the odds of knowledge of dispensed medication by 2.71 fold. The studies conducted in Gonder city and Chencha, Southwest Ethiopia also showed attending higher education was associated with increased odds of having sufficient knowledge of dispensed medication. The relationship may be due to the fact that educated patients can

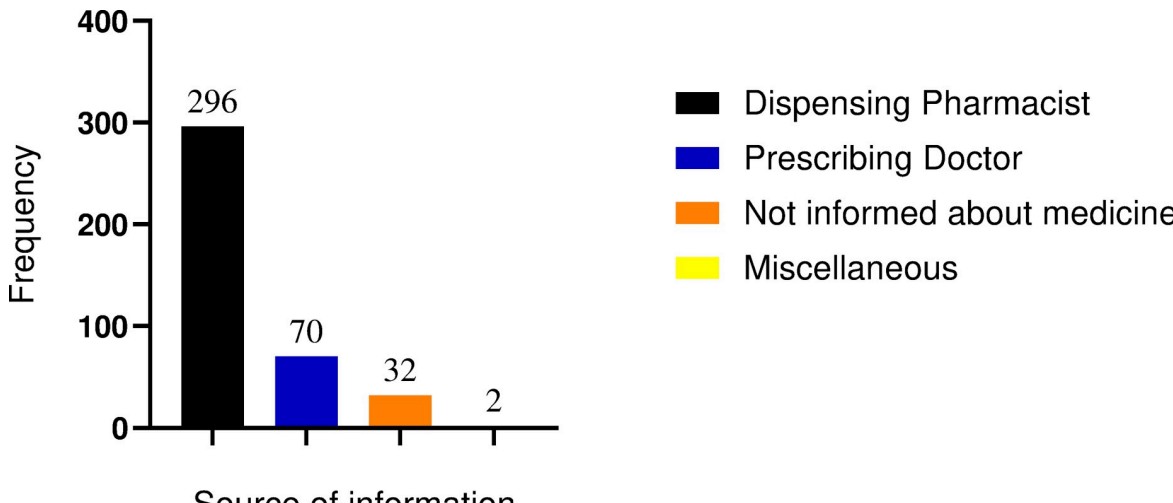

**Fig 1. Sources of drug information for ambulatory patients attending MTUTH from August to October 2021.**

**Table 4. Factors associated with exit-knowledge status of dispensed medications for ambulatory patients attending MTUTH from August to October 2021.**

| Variables | Sufficient Exit-Knowledge | | Odds ratio | | | |
|---|---|---|---|---|---|---|
| | Yes | No | COR (95% CI) | AOR (95% CI) | p-value | |
| | N (%) | N (%) | | | | |
| **Educational status** | | | | | | |
| Illiterate | 5(4.3%) | 31(10.9%) | 1.00 | 1.00 | | |
| Read and write | 5(4.3%) | 44(15.5%) | 0.592(0.154–2.277) | 0.590(0.113–3.086) | 0.532 | |
| Primary | 26(22.4%) | 71(25.0%) | 1.764 (0.552–5.639) | 1.336(0.326–5.478) | 0.687 | |
| Secondary | 32(27.6%) | 89(31.3%) | 1.312 (0.373–4.617) | 1.346(0.281–6.456) | 0.711 | |
| Tertiary | 48(41.4%) | 49(17.3%) | 3.787(0.99–14.485)* | 5.590(1.019–30.666) | **0.048** | |
| **Nature of disease** | | | | | | |
| Acute | 36(31.0%) | 199(70.1%) | 1.00 | 1.00 | | |
| Chronic | 80(69.0%) | 85(29.9%) | 5.941(3.625–9.735)* | 5.807(2.965–11.372) | **0.000** | |
| **Perceived sufficiency of pharmacist's information** | | | | | | |
| Enough | 95(81.9%) | 159(56.0%) | 1.00 | 1.00 | | |
| Do not know | 10(8.6%) | 54(19.0%) | 0.621(0.277–1.394)* | 0.374(0.142–0.982) | **0.046** | |
| Not enough | 11(9.5%) | 71(25.0%) | 0.361(0.179–0.730)* | 0.166(0.062–0.445) | **0.000** | |
| **Source of information** | | | | | | |
| Dispensing pharmacist | 70(60.3%) | 226(79.6%) | 1.00 | 1.00 | | |
| Prescribing doctor | 44(37.9%) | 26(9.2%) | 5.699(3.255–9.979)* | 7.620(3.608–16.093) | **0.000** | |
| Miscellaneous sources | 2(1.7%) | 0(0.0%) | 1.708(0.542–4.439) | 1.460(0.751–2.630) | 0.999 | |
| Not informed about the medication | 0(0.0%) | 32(11.3%) | 1.132 (0.733–3.617) | 1.432 (0.233–4.157) | 0.998 | |

*P≤0.05,

**P<0.001.

**Note**: Variables in bivariate analysis with p≤0.25 are indicated by *. Statistically significant in the multivariate analysis set at p≤ 0.05 set in **bold** typeface.

**Abbreviations:** AOR, adjusted odds ratio; CI, confidence interval; COR, crude odds ratio.

easily understand drug information provided by dispensers, and also they can easily obtain information about medicines from different sources.

The current study indicated that there was a significant association between patients' nature of disease and knowledge of dispensed medicines. Patients with chronic nature of diseases were more likely to have sufficient knowledge about dispensed medication than those with acute (AOR: 5.807; 95% CI 2.965–11.372). This finding is consistent with a study conducted in Sri Lanka [22]. This can be attributed to repeated exposure of chronic patients to information regarding their dispensed medications during follow-up. Furthermore, participants who claimed they "did not get enough information" and participants who "were not sure" to get sufficient information from the pharmacist had lower odds of 0.166 (95% CI 0.062–0.445) and 0.374 (95% CI: 0.142–0.982) of sufficient knowledge than those patients who responded "I got enough information from the pharmacist", respectively. This is expected that respondents who did not get enough information from the pharmacist would have insufficient exit knowledge regarding their dispensed medications. The source of information in our study was also significantly associated with patient knowledge of dispensed medicine. The probability of sufficiency of knowledge regarding dispensed medications was highest among patients who got information from prescribing doctors than dispensing pharmacists, miscellaneous sources, and not informed ones. Over-crowdedness of the dispensing environment, uncomfortability of the waiting area, and high workload of dispensers might be reasons for not getting enough information from them.

## Limitation of the study

This study has some limitations. The cross sectional and single centred nature of this study made it impossible to establish a causal relationship between the outcome variable and predictor variables, and infer the result to the larger population, respectively. Secondly, the patient's knowledge level was assessed at the exit of the outpatient pharmacy; this may not fully reflect how they use their medication at their home. Lastly, participants were considered knowledgeable when they responded a positive answer for at least two-thirds of the knowledge assessing questions; this might have resulted in an underestimation of patients' knowledge status. In addition to this, the face-to-face nature of the data collection can potentially lead to recall bias by respondents. Therefore, all these should be taken into consideration during the interpretation of the result.

## Conclusion

In the present study, the majority of patients seen at the outpatient pharmacy had insufficient exit knowledge about their dispensed medication. Based on the current study, it can be concluded that patients' educational level, perception of the sufficiency of obtained information, source of information, and nature of the disease has affected patients' exit-knowledge sufficiency. This overall insufficient patient knowledge of dispensed drugs has implications for treatment outcome and cost. Despite the presence or absence of other factors, pharmacists should provide good dispensing and counseling services to promote exit knowledge of patients about their dispensed medications.

## Acknowledgments

We would like to express our deepest gratitude to the study participants for their participation in this study. We are also grateful to our friends and family for their unreserved help, encouragement, and constructive comments.

## Author Contributions

**Conceptualization:** Semere Welday Kahssay.

**Data curation:** Semere Welday Kahssay.

**Formal analysis:** Semere Welday Kahssay, Peacock Mulugeta.

**Investigation:** Semere Welday Kahssay, Peacock Mulugeta.

**Methodology:** Semere Welday Kahssay, Peacock Mulugeta.

**Project administration:** Semere Welday Kahssay, Peacock Mulugeta.

**Resources:** Semere Welday Kahssay, Peacock Mulugeta.

**Software:** Semere Welday Kahssay, Peacock Mulugeta.

**Supervision:** Semere Welday Kahssay, Peacock Mulugeta.

**Validation:** Semere Welday Kahssay, Peacock Mulugeta.

**Visualization:** Semere Welday Kahssay, Peacock Mulugeta.

**Writing – original draft:** Semere Welday Kahssay, Peacock Mulugeta.

**Writing – review & editing:** Semere Welday Kahssay, Peacock Mulugeta.

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
