## [Decision Letter · Decision Letter 0]

11 Apr 2022

PONE-D-21-37588Exit-Knowledge About Dispensed Medications and Associated Factors Among Patients Attending the Outpatient Pharmacy of Mizan-Tepi University Teaching Hospital, Southern EthiopiaPLOS ONE

Dear Dr. Welday,

Thank you for submitting your manuscript to PLOS ONE. After careful consideration, we feel that it has merit but does not fully meet PLOS ONE’s publication criteria as it currently stands. Therefore, we invite you to submit a revised version of the manuscript that addresses the points raised during the review process.

We look forward to receiving your revised manuscript.

Kind regards,

Rafael Santos Santana

Academic Editor

PLOS ONE

Journal Requirements:

Additional Editor Comments:

The study has scientific merit and relevance to the research area. It should about changes pointed out by the reviewers.

Reviewers' comments:

Reviewer's Responses to Questions

**Comments to the Author**

1. Is the manuscript technically sound, and do the data support the conclusions?

Reviewer #1: Yes

Reviewer #2: Yes

2. Has the statistical analysis been performed appropriately and rigorously? 

Reviewer #1: Yes

Reviewer #2: Yes

3. Have the authors made all data underlying the findings in their manuscript fully available?

Reviewer #1: Yes

Reviewer #2: Yes

4. Is the manuscript presented in an intelligible fashion and written in standard English?

Reviewer #1: Yes

Reviewer #2: Yes

5. Review Comments to the Author

Reviewer #1: Patients and Methods:

In Study Population, what were the inclusion criteria?

What is the basis teorical for the decision to exclude patients with more than three dispensed medications?

Data Collection Techniques and Procedures, reference if the instrument used has been validated. How it's do calculation of knowledge level?

Ethical considerations, it is suggested to include the registration number with the Research and Ethical Review Committee of Mizan-Tepi University.

Result: Place each table in your manuscript file directly after the paragraph in which it is first cited (read order).

Discussion: To correct 177. This is not a prospective longitudinal study, as there is no follow-up of participants over time. I suggest that the discussion be revised, with the aim of bringing the reader alternatives to increase the level of output knowledge about drugs. I note that the study corroborates with several others, this is important for the text, however, I suggest that the discussion be expanded to how we can mitigate these factors associated with the lowest level of output knowledge about medicines. Thus, the study will give the reader something new, since similar analyzes and results have already been carried out before. I also suggest that the percentages previously presented in the results are not repeated in the discussion.

Reviewer #2: Exit-Knowledge About Dispensed Medications and Associated Factors Among Patients Attending the Outpatient Pharmacy of Mizan-Tepi University Teaching Hospital, Southern Ethiopia

Congratulations for this work

Indroducion

Line 41 suggest including as barriers "the high workload of dispensers” this barrier appears under discussion

Line 47 (…) halt the existence of drug therapy problems.

Are you sure about that? I suggest change the word (decrease?).

Line 59 (…) The findings of this study would promote good dispensing and counselling practices (…)

Could the findings promote good practices in dispensing and counseling? Consider this at the conclusion of the study.

Patients and Methods

Study Population

(…) Who fulfilled the inclusion criteria (…). What the inclusion criteria? Describe them.

(…) ambulatory patients visiting the outpatient pharmacy (…)

What patients? All patients? What the criteria did you use to select the patients?

Any patients refused to provide oral consent? If so, were excluded from the study?

Sample Size and Sampling Technique

Line 87 (…) eligible participants were consecutively enrolled until the required sample size reached (…)

include study eligibility criteria, because is not clear.

Data Collection Techniques and Procedures

Questionnaires were translated to local language?

Data Quality Control, Analysis and Interpretation

Line 105 (…) SPSS version 24

Include: The Statistical Package for the Social Science (SPSS) software version 24 for analysis…

Line 108 (…) Potential predictor variables with p value ≤0.25 in univariate analysis were considered for subsequent binary logistic regression analysis.

How did you choose the variables for this analysis (Table 4)? because in table 1 there are other variables that were not presented in table 4. For example: age group, sex, residence, etc., did they not influence the model?

Results

Line 133 (…) The main media of communication with the pharmacists was Benchigna.

It is not clear.

Patient- Pharmacist Interaction

Could the findings promote good practices in dispensing and counseling? Consider this at the conclusion of the study.

Discussion

Line 227 (...) This is obvious

Suggests change this word

The use of face-to-face interview can potentially lead to recall bias by respondentes, which should be taken into consideration while interpreting the result.

Table 1:

Exclude the symbol (%) from male and female variables

Conclusion: I would probably focus more on the implications of the findings about promote good practices in dispensing.

6. PLOS authors have the option to publish the peer review history of their article (what does this mean?). If published, this will include your full peer review and any attached files.

Reviewer #1: No

Reviewer #2: No

---

## [Author Response · Author response to Decision Letter 0]

19 Apr 2022

Response for reviewer -1

In Study Population, what were the inclusion criteria?

• Those ambulatory patients who were mentally stable, willing to participate, and visited the outpatient pharmacy during the study period were included in the study. (It is now included in the revised manuscript line 74).

What is the basis for the decision to exclude patients with more than three dispensed medications?

• The reason for excluding patients with more than three medications is that polypharmacy can potentially lead to increased difficulty in information retention and recalling ability; thus, it could negatively affect the overall study result.

reference if the instrument used has been validated.

• We used a questionnaire whose face validity was established using experts (senior staffs from our college) and it was also pilot tested on a subset of participants. In addition to these, principal components analysis (PCA) and Cronbach’s Alpha (CA) analysis were carried out and corrections were made accordingly.

How did you do the calculation of knowledge level?

• The participant was considered to have sufficient exit knowledge about the dispensed medication when he/she responded a positive answer for at least two-thirds (≥10 out of 15) of the total knowledge assessing questions. Those who gave a positive response for less than two-thirds (≤9 out of 15) of the total knowledge assessing questions were considered as having insufficient exit knowledge.

it is suggested to include the registration number with the Research and Ethical Review Committee of Mizan-Tepi University.

• Thank you for your suggestion, the registration number is now included in the revised manuscript (line 116).

Place each table in your manuscript file directly after the paragraph in which it is first cited (read order).

• Thank you, the tables are now placed after the paragraph they are first cited (general table guidelines are followed in the revised manuscript).

To correct 177. This is not a prospective longitudinal study, as there is no follow-up of participants over time. I also suggest that the percentages previously presented in the results are not repeated in the discussion.

• Thank you for your comments, but as mentioned earlier in the methodology section it was to mean a prospective cross-sectional study design, not a prospective cohort study design; thus, as you have stated there is no follow-up of participants over time. (the appropriate terminology is used in the revised manuscript line 198)

• The percentages of major findings are tried to be incorporated in the discussion section. 

Response for reviewer -2

Thank you for your comments and suggestions

Line 41 suggest including as barriers "the high workload of dispensers” this barrier appears under discussion

• Thank you, It is now included in the revised manuscript (line 43)

Line 47 (…) halt the existence of drug therapy problems. Are you sure about that? I suggest changing the word (decrease?).

• Thank you for your suggestion, the word change has been applied. (line 49)

Line 59 (…) The findings of this study would promote good dispensing and counselling practices (…)Could the findings promote good practices in dispensing and counseling? Consider this at the conclusion of the study.

• Yes it does, and now it is added in the conclusion of the revised manuscript (line 275)

(…) Who fulfilled the inclusion criteria (…). What the inclusion criteria? Describe them.

(…) ambulatory patients visiting the outpatient pharmacy (…)What patients? All patients? What the criteria did you use to select the patients?

• Those ambulatory patients who were mentally stable, willing to participate, and visited the outpatient pharmacy during the study period were included in the study. (now it is included in the revised manuscript).(line 74)

Any patients refused to provide oral consent? If so, were excluded from the study?

• No patients refused to provide oral consent.

Line 87 (…) eligible participants were consecutively enrolled until the required sample size reached (…) include study eligibility criteria, because is not clear.

• Thank you, now it is clearly stated in the revised manuscript.(line74) 

Questionnaires were translated into the local language?

• Yes, the questionnaire which was first prepared in English, was translated to the local language, Benchigna, to maintain consistency in translating the questionnaire (to keep it nuanced). (please see the revised manuscript line 103)

Line 105 (…) SPSS version 24 Include: The Statistical Package for the Social Science (SPSS) software version 24 for analysis

• Thank you, the full form is now included in the revised manuscript.(line 107)

Line 108 (…) Potential predictor variables with p-value ≤0.25 in univariate analysis were considered for subsequent binary logistic regression analysis. How did you choose the variables for this analysis (Table 4)? because in table 1 there are other variables that were not presented in table 4. For example: age group, sex, residence, etc., did they not influence the model?

• In table 4 only those predictor variables with P≤0.25 in univariate analysis are stated, and thus considered for subsequent binary logistic regression analysis. The predictor variables (like age, sex, residence…) were not included in table 4 because they were found to have a p-value >0.25 in univariate analysis; thus, not considered for bivariate analysis.

Line 133 (…) The main media of communication with the pharmacists was Benchigna. It is not clear.

• Benchigna, is a local language used by the communities where the study was conducted. The study revealed that, they communicate with the pharmacists mainly using this language. 

Could the findings promote good practices in dispensing and counseling? Consider this at the conclusion of the study.

• Yes, and now it is added to the conclusion of the revised manuscript.(line 275)

Line 227 (...) This is obvious. Suggests change this word

• Thank you for your suggestion, a word change has been applied (please see the revised manuscript line 249).

The use of face-to-face interviews can potentially lead to recall bias by respondents, which should be taken into consideration while interpreting the result.

• Thank you and it is now included in the limitation of the study and made clear for the readers (please see the revised manuscript line 266).

Table 1: Exclude the symbol (%) from male and female variables

• Thank you. Now it is cleared

---

## [Editor Report · Decision Letter 1]

12 May 2022

Determinants of exit-knowledge of ambulatory patients on their dispensed medications: the case in the outpatient pharmacy of Mizan-Tepi University Teaching Hospital, Southwest Ethiopia

PONE-D-21-37588R1

Dear Dr. Semere Welday,

We’re pleased to inform you that your manuscript has been judged scientifically suitable for publication and will be formally accepted for publication once it meets all outstanding technical requirements.

Kind regards,

Rafael Santos Santana

Academic Editor

PLOS ONE

---

## [Editor Report · Acceptance letter]

13 May 2022

PONE-D-21-37588R1 

Determinants of exit-knowledge of ambulatory patients on their dispensed medications: the case in the outpatient pharmacy of Mizan-Tepi University Teaching Hospital, Southwest Ethiopia 

Dear Dr. Welday Kahssay:

I'm pleased to inform you that your manuscript has been deemed suitable for publication in PLOS ONE. Congratulations! Your manuscript is now with our production department. 

Kind regards, 

on behalf of

Dr. Rafael Santos Santana 

Academic Editor

PLOS ONE